# EMERGENT LANGUAGE-BASED DIALOG FOR COLLABORATIVE MULTI-AGENT NAVIGATION

## ABSTRACT

This paper aims to provide an empirical study on how to build agents that can collaborate effectively with multi-turn emergent dialogues. Recent research on emergent language for multi-agent communications mainly focuses on single-turn dialogue and simple settings where observations are static during communications. Here, we propose a complicated yet more practical multi-agent navigation task: the Tourist (the embodied agent) who can observe its local visual surroundings, and the Guide who has a holistic view of the environment but no foreknowledge of the Tourist's location. The objective of the task is to guide the Tourist to reach the target place via multi-turn dialogues emerging from scratch. To this end, we propose a collaborative multi-agent reinforcement learning method that enables both agents to generate and understand emergent language, and develop optimal dialogue decisions with a long-term goal of solving the task. We also design a real-world navigation scene with matterport3D simulator (Anderson et al. (2018)). The result shows that our proposed method highly aligns emergent messages with both surroundings and dialogue goals, hinting that even though without human annotation or initial meaning, the agents can learn to converse and collaborate under task-oriented goals [1].

## 1 INTRODUCTION

As artificial agents become more effective at solving specialized tasks, there has been growing interest in emergent communication (EC) that enables multi-agents to flexibly interact to address joint tasks, recent works have proven that neural agents can learn to develop interaction languages and communicate using them from scratch without external supervision when they are trained to cooperate (Lazaridou et al. (2018); Chaabouni et al. (2019); Kharitonov & Baroni (2020); Ren et al. (2020); Mu & Goodman (2021); Tucker et al. (2022)) in virtual games such as the Lewis signaling game (Lazaridou et al. (2016), negotiation game Cao et al. (2018)), circular biased game (Noukhovitch et al. (2021)), and so on.

Most of these works explore a single-turn interaction between two agents, which models a unidirectional message-passing process. However, most real-world tasks pose a necessary bidirectional message passing where all involved agents are capable of generating responses whilst understanding a long-horizon dialogue, and of aligning the messages with real-world surroundings and dialogue intents. Besides, most existing works assume a static conversation environment where the observations about the environment keep constant during communication. Consequently, the involved agents are disabled to conceive the environment in a functional and active manner.

In this work, we aim to address these issues by linking emergent language with more complicated yet practical tasks. Considering navigation is a fundamental but critical ability of agents interacting with the real world dynamically, we propose a collaborative multi-agent navigation task by exploring interactions via emergent language, inspired by Vision-and-Language Navigation (VLN) task (Anderson et al. (2018)) with the purpose of building visual agents that follow natural language instructions and navigate in real scenes. This task involves two agents: the Tourist (the embodied agent) who possesses the ability to observe its immediate surroundings and make decisions to move to the subsequent location, and the Guide (serving as the oracle agent in the VLN task) who main-

---

[1]We will open-source the simulation suite, algorithm implementation, and pre-trained model when the paper is accepted.

tains a comprehensive overview of the entire environment and is aware of a predetermined target location, yet lacks any knowledge regarding the current whereabouts of the Tourist. With the goal of guiding the Tourist to the target place, the Guide should guess where the Tourist is located and generate navigation instructions. Simultaneously, the Tourist should comprehend these instructions and progressively send messages describing its visual observations to the Guide.

In order to learn autonomous and more functionally efficient communication, we propose a multi-agent collaborative reinforcement learning (RL) framework. Within this framework, the language generating and language understanding modules can be simultaneously trained, with successfully solving the navigation task in the fewest possible dialogue turns as the core reward. Moreover, in order to alleviate the instability of the end-to-end RL training, and meanwhile to learn better representations of complicated observations, we put forth two auxiliary learning tasks: guessing the location of the Tourist and following the instructions of the Guide. Such tasks involve single-step "speak and listen" based on image classification and instruction-trajectory matching. We empirically show that the two auxiliary learning tasks substantially contribute to training the emergent communication model with RL. We carry out extensive experiments in various settings with different vocabulary sizes, variant message lengths, and varying exploring strategies. The proposed methods yield consistently promising results, even when benchmarked against models trained with human-labeled data and foreknowledge of the intricate aspects of languages.

We summarize our contributions as follows:

1. We propose a new multi-agent emergent communication task that features a real-world navigation scene, necessitating agents to develop emergent language and engage in continuous interactions amidst a dynamic environment.

2. We establish a novel multi-agent reinforcement learning (RL) framework complemented by auxiliary pre-training to effectively align emergent language with both the environment and the task .

3. We provide an empirical study about building agents to solve real-world problems with multi-turn emergent dialogues, and an intensive study about the effect of different vocabulary sizes, variant message lengths, and distinct dialogue strategies.

## 2 RELATED WORK

**Emergent Communication.**

The majority of existing emergent communication tasks stem from the Lewis signaling game (also known as the referential game) Lazaridou et al. (2016); Evtimova et al. (2017); Chaabouni et al. (2020; 2022); Xu et al. (2022); Rita et al. (2022); Tucker et al. (2022), which requires the speaker to convey information about a target image using emergent language, while the listener endeavors to deduce the target image from a pool of candidate images. Additionally, there are several works proposing variations of emergent communication tasks. Noukhovitch et al. (2021) proposes a competitive task where the sender and the receiver have their own target place and they try to let the sender choose the position close to their own targets. Cao et al. (2018) proposes a negotiation task that requires the agents to reach an agreement about the distribution of the objects by maximizing their own interests.

There are some efforts on the navigation task in the emergent communication area. de Vries et al. (2018) has a similar setting to our work but lets the Tourist describe the its current observation in emergent language, and the Tourist (the embodied agent) walks randomly until the Guide predicts its location as the target. Since there is no exchange of information that enables them to coordinate, it leads to poor performance. Kajic et al. (2020); Kalinowska et al. (2022) design a navigation task in a limited number of grid-world environments. While closely related to our communication architecture, these methods only consider the simple alignment of language and simulated structures, which quickly becomes intractable in real-world tasks with high-dimensional observation spaces and long time horizons. Patel et al. (2021) explores the communication in a more realistic environment but it passes continuous vectors with simple communication module.

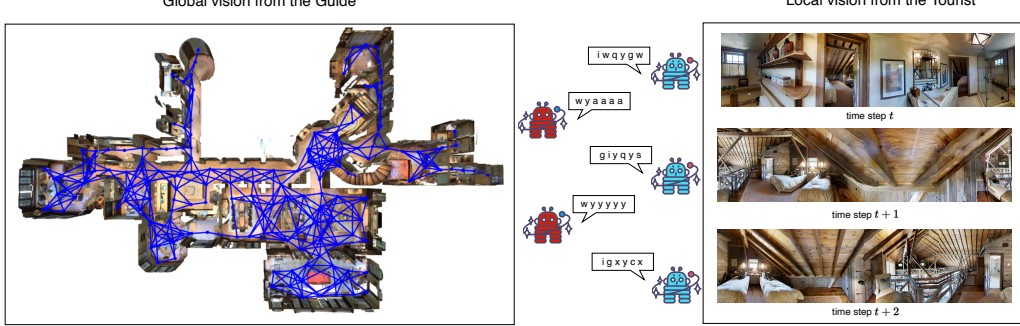

Figure 1: Navigation game adapted from Anderson et al. (2018).

**Vision-and-Language Navigation.** Our work is also related to recent work on vision-and-language (VLN) navigation whose aim is to train an embodied agent to follow the given natural language instructions and arrive at the target place. Anderson et al. (2018) first proposes the VLN task, and provides a sequence-to-sequence baseline using the LSTM model. Some prominent researches focus on exploring more efficient learning architecture (Landi et al. (2019), Li et al. (2019), Hong et al. (2020), Chen et al. (2021), Chen et al. (2022)), with the assumption that the high-level instruction is provided at the beginning and no information exchanging is needed in following procedure.

There are also works training the embodied agent to navigate by interpreting the natural language from the oracle agent. Thomason et al. (2020) and Padmakumar et al. (2022) propose navigation from dialog history task, requiring the embodied agent to navigate to the target place with its dialogue history with the oracle. Nguyen & Daumé III (2019) and Nguyen et al. (2019) propose tasks that allow the embodied agent to ask for help from the oracle. Roman et al. (2020) propose to model both of the embodied agent and the oracle to complete the navigation task.

Different from the works above, firstly, we focus on the emergent communication ability of the agents, other than using natural language. Secondly, we enable the embodied agent not only to understand language but also generate language, rather than one of them, and we also empower the oracle agent to provide instructions progressively, in contrast to relying on fixed dialogue histories or rule-based simulators. Thirdly, the agents are trained to engage in dialogue through emergent language from scratch without using any annotated data or any knowledge of the language.

## 3  TASK DEFINITION

As illustrated in Figure 1, the multi-agent navigation task consists of two agents: the Guide and the Tourist. It requires the Guide to give walking instructions by inferring the Tourist's current location, and the Tourist, to dialog with the Guide and follow its instructions to reach the goal location. The involved two agents learn to communicate through unsupervised emergent language.

Formally, at the time step $t$ in an episode, the Guide, equipped with a global environment represented as a weighted, undirected graph $G = \langle V, E \rangle$, is required to give helpful instructions to the Tourist as soon as it receives message $U_t^T$ from the Tourist. Given the dialog history $H_{dialog}^G = \langle U_1^T, U_1^G, U_2^T, U_2^G, ..., U_{t-1}^T, U_{t-1}^G, U_t^T \rangle$, and the representation of the $G = \langle V, E \rangle$, the Guide first predicts the Tourist's location, and then generates message $U_t^G = \langle u_1^G, u_2^G, ..., u_{L^u}^G \rangle$ to describe the movement instructions that the Tourist is expected to take, or to ask for more information to help the Guide to locate where the Tourist is, where $L^u$ is the maximum length of the utterance and $u_i \in Voc$ is a word token coming from the pre-defined vocabulary of emergent language. All tokens have no initial meanings before the training process. $V = \langle o_{pos_1}, o_{pos_2}, ..., o_{pos_K} \rangle$ is the image observation of every position and $E$ represents the connection relationship among those positions which is used to get the shortest path from any position $pos_i$ to the target position $tgt$, $P^i = ShortestPath(E, pos_i, tgt) = \langle a_1^{pos_i \to pos_j}, a_2^{pos_j \to pos_k}, ..., a_{L^p}^{pos_m \to tgt} \rangle$, where $L^p$ is the length of the shortest path.

The Tourist is required to follow instructions from the Guide to make an action, it could choose to stay where it is or move to the reachable positions within one step[2]. It was given as input all previously generated dialogues in emergent language $H_{dialog}^T = \langle U_1^T, U_1^G, U_2^T, U_2^G, ..., U_{t-1}^T, U_{t-1}^G \rangle$,

---

[2]We use letter indexing "T" for the Tourist and "G" the Guide.

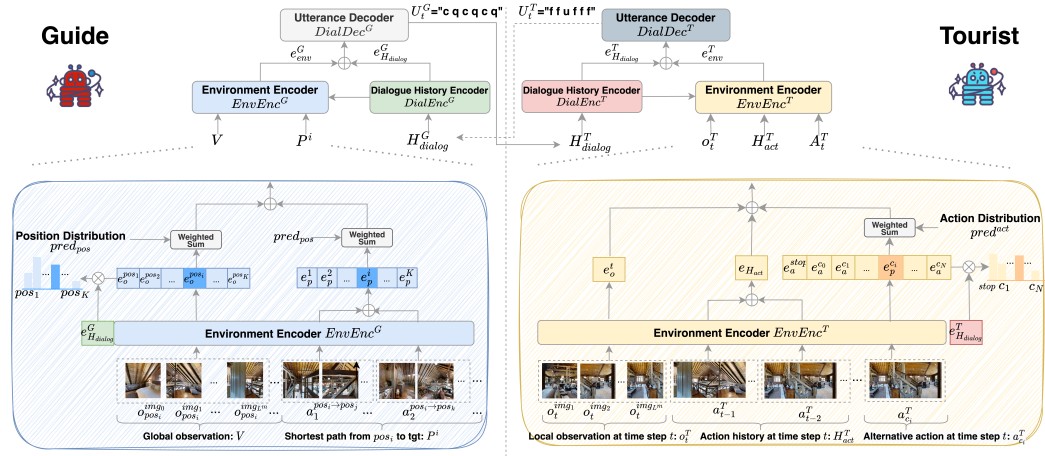

Figure 2: The framework. Upper: the Guide and the Tourist are trained jointly under the full reinforcement learning with planning over long time horizons. Lower: two auxiliary tasks that involve single-step "speak and listen" based on image classification and instruction-trajectory matching.

its action history $H_{act}^T = \langle a_1^T, a_2^T, ..., a_{t-1}^T \rangle$, and its current visual observation $o_t^T = \langle o_t^{img_1}, o_t^{img_2}, ..., o_t^{img_{L^m}} \rangle$ with $L^m$ images, to predict its next actions $a_t^T$ from candidate actions $A_t^T = \langle a_{stop}^T, a_{c_1}^T, a_{c_2}^T, .., a_{c_N}^T \rangle$, where $a_{stop}^T$ represents staying in the same position and $a_{c_i}^T$ means moving to the neighbor position $c_i$. $a_{ci}^T = \langle a_i^{img_1}, a_i^{img_2}, ..., a_i^{img_{L^a}} \rangle$ is represented with the observed $L^a$ images in the angles from which the Tourist can move to the neighbor position $c_i$. After the Tourist takes an action $a_t^T$, it then generates and sends out a message $U_t^T$ to describe its new observations or to further inquiry for its next action.

## 4 METHODOLOGY

As illustrated in Figure 2, the two agents perceive both environmental information and contextual dialogues, and generate interactive messages to exchange information and collaboratively fulfill the task.

### 4.1 ENCODER OF THE GUIDE AGENT

**Dialogue Encoding.** A transformer encoder $DialEnc^G$ takes all the history dialogues $H_{dialog}^G$ between the Guide and the Tourist, and obtains contextual representation:

$$e_{H_{dialog}}^G = DialEnc^G(H_{dialog}^G) \in \mathbb{R}^{L^d \times dim} \tag{1}$$

**Environment Encoding.** The environment encoder $EnvEnc^G$ applies the $ResNet$ to encode the global visions of all $K$ positions, with each consisting of $L^m$ images, $o_{pos_i} = \langle o_{pos_i}^{img_1}, o_{pos_i}^{img_2}, ..., o_{pos_i}^{img_{L^m}} \rangle$, and obtains the global environment representation $e_{res}^{pos} = (e_{res}^{pos_1}, ..., e_{res}^{pos_i}, ..., e_{res}^{pos_K})$, with each $e_{res}^{pos_i}$ calculated as:

$$e_{res}^{pos_i} = ResNet(o_{pos_i}) \in \mathbb{R}^{L^m \times dim_{res}} \tag{2}$$

then the concatenation of them is fed into a transformer encoder to capture position information and a trainable environment encoder:

$$e_o^{pos_i} = EnvEnc^G(e_{res}^{pos_i}) \in \mathbb{R}^{L^m \times dim} \tag{3}$$

To predict the current position of the Tourist, the Guide calculates the distribution on all $K$ candidate positions with the hint of $e_{H_{dialog}}^G$:

$$pred^{pos_i} = \frac{exp(\bar{e}_o^{pos_i} \cdot \bar{e}_{H_{dialog}}^G)}{\sum_j exp(\bar{e}_o^{pos_j} \cdot \bar{e}_{H_{dialog}}^G)} \in \mathbb{R}, \tag{4}$$

$$e_o^{pos} = \sum_{i=1}^{K} pred^{pos_i} e_o^{pos_i} \in \mathbb{R}^{L^m \times dim}, \tag{5}$$

where $\bar{e}_o^{pos_i}$ and $\bar{e}_{H_{dialog}}^G$ are mean-pooling of the original matrix $e_o^{pos_i}$ and $e_{H_{dialog}}^G$, respectively. In this way, $e_o^{pos}$ is deemed to capture the current Guide position in a global environment.

In order to generate helpful walk instruction, the Guide should be capable of planing the shortest route $P^i$ for each position $pos_i$ to the target place $tgt$. Similar to the global vision encoding, a $ResNet$ is used to encode each path $pos_j$ to $pos_k$:

$$e_{res}^{pos_j \to pos_k} = ResNet(a_1^{pos_j \to pos_k}), \tag{6}$$

$$e_{res}^{P_i} = concat(e_{res}^{pos_i \to pos_j}, e_{res}^{pos_j \to pos_k}, ..., e_{res}^{pos_m \to tgt}), \tag{7}$$

$$e_p^i = EnvEnc^G(e_{res}^{P_i}) \in \mathbb{R}^{(L^p \times L^a) \times dim}. \tag{8}$$

The planed route is a weighted sum of the predicted position distribution $e_{res}^{pos_i}$ and the global path $e_p^i$:

$$e_p = \sum_{i=1}^{K} pred^{pos_i} e_p^i \in \mathbb{R}^{(L^P \times L^a) \times dim} \tag{9}$$

Then, the environment encoding is obtained by concatenating the position information and the route information:

$$e_{Env}^G = Concate(e_o^{pos}, e_p). \tag{10}$$

Finally, the encoded dialogue history and the encoded environment information are concatenated as the encoder output of the Guide:

$$e^G = Concat(e_{H_{dialog}}^G, e_{Env}^G) \in \mathbb{R}^{(L^d + L^m + L^p \times L^a) \times dim}. \tag{11}$$

## 4.2 DECODER OF THE GUIDE AGENT

With the encoder's output $e^G$ as input, a transformer decoder $DialDec^G$ is applied to generate instruction message for the Tourist:

$$U_t^G = DialDec^G(e^G) \tag{12}$$

## 4.3 ENCODER OF THE TOURIST AGENT

**Dialogue Encoding.** The Tourist also utilizes a transformer encoder $DialEnc^T$ to summarize the dialogue history $H_{dialog}^T$:

$$e_{H_{dialog}}^T = DialEnc^T(H_{dialog}^T) \in \mathbb{R}^{L^d \times dim}. \tag{13}$$

**Environment Encoding.** Different with the environment of the Guide, the Tourist takes its current visual observations $o_t^T$, its history actions $H_{act}^T$, and the candidate action $a_{c_i}^T$ in current position as its local environment. In a manner akin to the environment encoder of the Guide, the three visual variables are fed into a $ResNet$ and then a transformer encoder $EnvEnc^T$, respectively, yielding the corresponding encoding $e_o^t \in \mathbb{R}^{L^m \times dim}$, $e_{H_{act}} \in \mathbb{R}^{((t-1) \times L^a) \times dim}$, and $e_a^{c_i} \in \mathbb{R}^{L^a \times dim}$.

The next action distribution $pred^{act} = \langle pred^{act_{stop}}, pred^{act_1}, pred^{act_2}, ..., pred^{act_N} \rangle$, with each $pred^{act_i}$ calculated as under the pilot of contextual dialogue $e_{H_{dialog}}^T$:

$$pred^{act_i} = \frac{exp(\bar{e}_{H_{dialog}}^T \cdot \bar{e}_a^{c_i})}{\sum_j exp(\bar{e}_{H_{dialog}}^T \cdot \bar{e}_a^{c_j})}, \tag{14}$$

$$e_a = \sum_{i=1}^{N+1} pred^{act_i} e_a^{c_i} \in \mathbb{R}^{L^a \times dim}, \tag{15}$$

where $\bar{e}_a^{c_i}$ and $\bar{e}_{H_{dialog}}^T$ are mean-pooling of the original matrix $e_a^{c_i}$ and $e_{H_{dialog}}^T$, respectively. In this way, $e_a$ is deemed to capture the current movement decision.

Then, the environment encoding of the Tourist is captured by $e_{env}^T = Concat(e_o^t, e_{H_{act}}, e_a)$.

Finally, the encoded dialogue history and the encoded environment information are concatenated as the encoder output of the Tourist:

$$e^T = Concat(e_{H_{dialog}}^T, e_{env}^T) \in \mathbb{R}^{(L^d + L^m + t - 1 + L^a) \times dim}. \tag{16}$$

## 4.4 Decoder of the Tourist Agent

Similar to the Guide's decoder, the Tourist also uses a transformer decoder $DialDec^T$ to generates message describing its current state:

$$U_t^T = DialDec^T(e^T). \tag{17}$$

## 4.5 Training Setup

The model outlined in Figure 2 is trained under a reinforcement learning (RL) in an end-to-end manner. However, the RL algorithm struggles in solving long-horizon navigation task with sparse rewards. To this end, we design two pre-training tasks by spitting the navigation task into two steps: localization and movement.

**Localization.** In the localization task, the Guide predicts the Tourist location and compares it with the ground-truth location. In this end, $DialDec^T$ and $EnvEnc^G$ are trained with REINFORCE algorithm, with the reward to be +1 if the Guide makes a right prediction, and -1 otherwise:

$$\nabla \mathcal{L}_{loc} = E[\frac{1}{L^d} \sum_t R_{loc}^t(\cdot) \nabla log(\pi(U_t^T | S_{loc_t}^T)) + R_{loc}^{t+1}(\cdot) \nabla log(\pi(U_t^G | S_{loc_t}^G))] \tag{18}$$

where $S_{loc_t}^T = \langle H_{dialog}^T, o_t^T \rangle$ and $S_{loc_t}^G = \langle H_{dialog}^G, Graph \rangle$. The localization task facilitates that the Tourist generates message aligned with its environment, meanwhile the Guide comprehends the true meaning the message from the Tourist.

**Movement.** In the movement task, the Tourist takes action (stops or moves) and compares it with the ground truth route instruction from the Guide. In this end, $DialDec^G$ and $EnvEnc^T$ are trained with REINFORCE algorithm, with the reward to be +1 if the Tourist takes a right action, and -1 otherwise:

$$\nabla \mathcal{L}_{move} = E[\frac{1}{L^d} \sum_t R_{move}^{t+1}(\cdot) \nabla log(\pi(U_t^G | S_{move_t}^G)) + R_{move}^{t+1}(\cdot) \nabla log(\pi(U_t^T | S_{move_t}^T))], \tag{19}$$

where $S_{move_t}^T = \langle H_{dialog}^T, H_{act}^T, o_t^T \rangle$ and $S_{move_t}^G = \langle H_{dialog}^G, Graph, pos^T, tgt \rangle$. The movement task facilitates that the Guide clearly describes its instruction and the Tourist accurately understands the meaning of message from the Guide.

**Navigation Task.** In the navigation task, the core modules are trained together with REINFORCE algorithm, with the summed rewards of the localization and the movement:

$$\nabla \mathcal{L}_{nav} = E[\frac{1}{L^d} \sum_t (R_{loc}^{t+1}(\cdot) + R_{move}^{t+1}(\cdot)) \nabla log(\pi(U_t^G | S_{nav_t}^G))$$
$$+ (R_{loc}^t(\cdot) + R_{move}^{t+1}(\cdot)) \nabla log(\pi(U_t^T | S_{nav_t}^T))], \tag{20}$$

where $S_{nav_t}^T = \langle H_{dialog}^T, H_{act}^T, o_t^T \rangle$ and $S_{nav_t}^T = \langle H_{dilog}^G, Graph, tgt \rangle$.

## 5 Experiments

### 5.1 Experiment Setup

**Dataset.** We evaluate our method on R2R Anderson et al. (2018) and CVDN Thomason et al. (2020), both of which are created for the visual-language navigation task. The R2R is built upon the Matterport3D simulator and collects 7,189 paths in 90 houses with 10,567 panoramas. Each path has 3 initial instructions. It requires the agent to navigate to the target place given the initial natural language instructions. The CVDN dataset is also built upon the Matterport3D simulator, having 7k trajectories and 2,050 navigation dialogs. It requires the agent to navigate given the dialog history. Both of the datasets are split into train, val seen, val unseen, and test datasets. It is worth noting that, with the aim of building an emergent language system, we only utilize the simulated environment of the two datasets, and discard all natural language dialogues. The R2R and CVDN datasets have their leaderboards, and test results are demonstrated on it with the ground truth unknown to the public. Since our work outputs emergent language other than natural language, the evaluation of the test dataset cannot be conducted as the VLN leaderboard requires. In the training process on the R2R,

| Method | Val Seen | | | | Val Unseen | | | |
|---|---|---|---|---|---|---|---|---|
| | TL | NE ↓ | SR ↑ | SPL ↑ | TL | NE ↓ | SR ↑ | SPL ↑ |
| Seq2Seq (2018) | 11.33 | 6.01 | 39 | - | 8.39 | 7.81 | 22 | - |
| EnvDrop (2019) | 11.00 | 3.99 | 62 | 59 | 10.70 | 5.22 | 52 | 48 |
| TDSTP (2022) | - | 2.34 | 77 | **73** | - | 3.22 | 70 | 63 |
| HAMT (2021) | 11.15 | 2.51 | 76 | 72 | 11.46 | 3.62 | 66 | 61 |
| ESceme (2023) | 10.65 | 2.57 | 76 | **73** | 10.80 | 3.90 | 68 | 64 |
| Ours (Navigation) | 12.57 | **1.74** | **78** | 67 | 12.22 | **1.69** | **77** | **65** |

Table 1: Comparison with the prominent works on R2R dataset.

| Category | Method | Goal Progress ↑ | | Adapted Goal Progress ↑ | |
|---|---|---|---|---|---|
| | | Val Seen | Val Unseen | Val Seen | Val Unseen |
| Single | Seq2Seq(2018) | 5.92 | 2.10 | - | - |
| Single | MT-RCM+EnvAg(2020) | 5.07 | 4.65 | - | - |
| Single | HAMT(2021) | 6.91 | 5.12 | - | - |
| Single | ESceme (2023) | **8.34** | 5.42 | - | - |
| Multiple | RMM(2020) | - | - | **15.0** | 5.6 |
| Multiple | Ours (Navigation) | 7.13 | **7.07** | 13.06 | **11.38** |

Table 2: Comparison with the existing works on CVDN dataset. "Single" means the method only models the Tourist agent, while "Multiple" means it models both of the agents.

we choose the start and the destination of the Tourist by restricting their shortest path within 4 to 6 hops as the most existing works on the VLN task.

**Evaluation Metrics.** We adopt 6 metrics to evaluate the performance of our method. (1) Trajectory Length (TL): the average length of the Tourist navigated route. (2) Navigation Error (NE): the distance between the Tourist's final position and the target place. (3) Success Rate (SR): a trajectory is counted to be successful only if the shortest path between the final position and the target place is less than 3 meters, and the success rate is the ratio of successful trajectory. (4) Success Rate weighted by Path Length (SPL): suppose $S_i = 1$ when the episode $i$ succeeds, and 0 when it fails, $l_i$ is the shortest path distance from the start position to the target, $p_i$ is the navigated path distance of the Tourist, then SPL is calculated as the equation: $\frac{1}{I} \sum_i^T S_i \frac{l_i}{\max(l_i, p_i)}$. (5) Goal Progress (GP): Given a dialogue history, it measures the difference between the shortest path length from the current position (the position where the Tourist has a conversation with the Guide in the CVDN dataset) and that from the reached position to the target place. (6) Adapted Goal Progress: It measures the difference between the shortest path length from the start position and that from the reached position to the target place.

**Implementation Details.** The layers of all the Transformer encoders and decoders are set to 3. ResNet-18 (He et al. (2016)) is used to encode the visuals and is frozen during training. The vocabulary size is set to 26. The learning rate is set to be 0.001, the batch size 50, and the maximum time step to be 10 on the R2R, and 20 on the CVDN. As in the works of VLN, each position observes 36 images ($L^m = 36$).

## 5.2 PERFORMANCE ON THE R2R DATASET

To draw a paralleled comparison with related works on the R2R dataset, the Guide is provided with the Tourist's position only at the beginning. In the following steps, it has to speculate on where the Tourist is according to the message from the Tourist. The results in Table 1 indicate that our method has achieved the lowest navigation error (NE) and a compelling success rate (SR) in val seen. Such advantages are more significant, particularly in the val unseen. But it yields longer trajectories (TL), with the most likely reason being that it makes more attempts to exchange information during navigating.

It is noteworthy that our work distinguishes itself from all the comparisons by modeling both the embodied agent (the Tourist) and the oracle agent (the Guide), which is more challenging. Moreover, our method simultaneously equips the involved agents to both comprehend and generate the language, without relying on rule-based simulators or any fixed contexts as that in the related VLN works. Most importantly, our method stands out for its lack of dependency on any human-annotated information.

| Message | Localization | Movement | | | | Navigation | | | |
|---|---|---|---|---|---|---|---|---|---|
| Length | Acc ↑ | TL | NE ↓ | SR ↑ | SPL ↑ | TL | NE ↓ | SR ↑ | SPL ↑ |
| L=0 | 1.16 | 21.23 | 9.31 | 6.62 | 2.80 | 21.23 | 9.31 | 6.62 | 2.80 |
| L = 1 | 52.28 | 13.05 | 1.35 | 81.87 | 62.22 | 18.29 | 8.18 | 11.62 | 5.71 |
| L = 3 | 64.99 | 10.72 | 0.05 | 99.37 | 94.28 | 17.01 | 5.06 | 40.37 | 26.07 |
| L = 6 | **68.86** | 10.37 | **0.05** | **99.50** | 93.96 | 14.91 | 3.44 | 56.62 | 41.64 |
| L = 12 | 57.00 | 10.63 | 0.10 | 99.12 | **94.17** | 12.90 | **1.94** | **74.75** | **61.31** |
| Virable L | 41.51 | 9.71 | 1.51 | 80.80 | 74.70 | 14.01 | 6.63 | 25.37 | 16.96 |

Table 3: Results with different settings of message length.

| Dialogue | Movement | | | | Navigation | | | |
|---|---|---|---|---|---|---|---|---|
| Frequency | TL | NE ↓ | SR ↑ | SPL ↑ | TL | NE ↓ | SR ↑ | SPL ↑ |
| w/o dialogue | 21.23 | 9.31 | 6.62 | 2.80 | 21.23 | 9.31 | 6.62 | 2.80 |
| only in beginning | 10.79 | 6.44 | 25.37 | 21.13 | 9.99 | 7.41 | 22.62 | 19.69 |
| soft | 11.72 | 6.89 | 28.74 | 23.28 | 10.95 | 7.25 | 24.62 | 20.20 |
| 20% | 14.85 | 5.57 | 37.25 | 27.45 | 14.73 | 6.05 | 37.37 | 27.20 |
| 50% | 15.26 | 3.56 | 62.87 | 45.64 | 16.41 | 5.43 | 43.25 | 28.14 |
| 100% | 10.63 | **0.10** | **99.12** | **94.17** | 12.9 | **1.94** | **74.75** | **61.31** |

Table 4: Results on different settings of dialogue strategy.

## 5.3 PERFORMANCE ON THE CVDN DATASET

We adopt the same settings of targets and the length of the planner path as in the related works on the CVDN dataset. One difference is that the Guide in our method has to guess where the Tourist is at each time step, other than being provided with the correct locations of the Tourist. The result in Table 2 demonstrates that our method achieves significantly higher goal progress and adapted goal progress in the val unseen datasets, but lower results in the val seen ones. It signifies that the emergent language-based dialogue system generalizes eminently on unseen scenes, even trained from scratch without any prior knowledge of language and task.

## 5.4 ABLATION STUDIES

**Different Message Length.** To investigate the effect of message length on performance, we conduct experiments by fixing the message length to 0 (w/o communication), 1, 3, 6, and 12. As depicted in Table3, the performance steadily increases as the message length tunes from 0 to 6, but there is a decline when the message length increases to 12, despite the fact that longer messages tend to provide more expression space. The most likely reason is that the action space of the decoder grows exponentially with the message length increase, making it harder to align the symbol sequence with specific dialogue intents. As opposed to a fixed length, we further investigate the scenario of variable sentence lengths, wherein the agent decides and generates messages within 20 symbols. Our experiments observe that the agents tend to greedily generate messages with the full 20 symbols even though with a penalty for a longer sentence, which inspires in-depth future investigation.

**Different Dialogue Strategies.** To investigate the importance of exchanging information, we examine the effect of different dialogue strategies: (1) w/o dialogue: at each step the Tourist walks randomly without any communication with the Guide. (2) only in beginning: The Guide gives walking instructions at the beginning, with no further communication thereafter. (3) soft: at each step the Tourist decides whether to communicate with the Guide according to the belief of its movement action. We determine the degree of belief through a straightforward criterion: if the score gap of the top two actions is lower than a threshold $\rho$, then the Tourist is deemed to be confused about the optimal movement and therefore initiates communication with the Guide. (4) 20%: The Tourist communicates with the Guide once every 5 turns throughout the whole episode. (5) 50%: once every 2 turns. (6) 100%: The Tourist communicates with the Guide at every turn. As shown in Table 4, the performance improves as the dialogue frequency grows. From the view of cooperation, more dialogues facilitate the exchange of additional information regarding the environment and the task at hand, yielding better performances. From the view of training, more dialogues bring more chances for the model to optimize, leading to formulate a language with higher quality. In our experiment, the average dialogue frequency in the soft setting is 15.22%, lower than 20%, yielding a poorer performance. Although with the intuition that better task performance with lower SPL leads to shorter trajectory lengths, we observe an upward trend in trajectory lengths when dialogue frequency goes up from "only in beginning" to "50%". Through intensive case studies, we find that the Tourist tends to randomly move with less instructions, but tends to stop moving if it made wrong movements in the last few steps and wait for more information from the Guide to encourage itself move

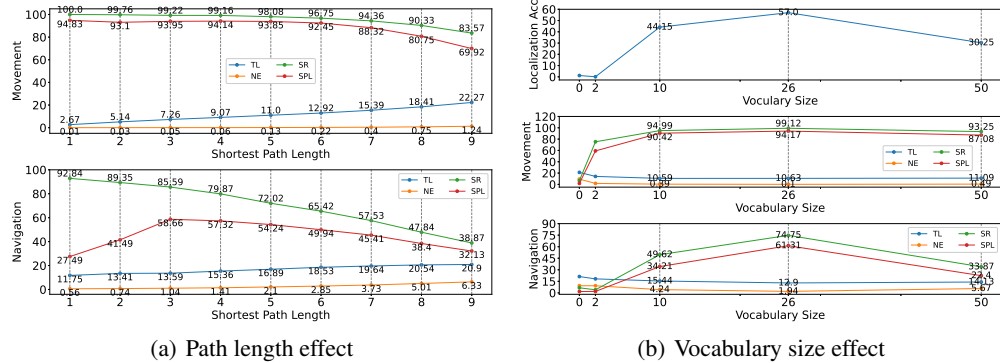

| (a) Path length effect | (b) Vocabulary size effect |

Figure 3: (a) Results with different shortest path length between start position and destination. (b) Results with different settings of vocabulary size.

again, till the dialogue terminates at the maximum time step, which explains the rising of the TL when dialogue frequency varying from "only in beginning" to "50%". Higher dialogue frequency helps accumulating more information about movement, therefore, the TL decreases when dialogue frequency set to be "100%". Since wrong movements make the path to the destination much longer, the "100%" case with higher one-step movement accuracy has a shorter path.

**Varying Vocabulary Size.** In order to analyze the effect of vocabulary size, we conduct experiments by varying the vocabulary size 0, 2, 10, 26, and 50 respectively. As shown in Figure 3(b), the performance does not grow as the vocabulary grows, which again confirms our statement that despite the growth of vocabulary size enriches the space of expression, it expands the action space which makes it much harder to learn a high-quality expression, similar to the effect of the message length.

**Varying Task Difficulty.** We enhance the level of task complexity by assessing the number of hops between the initial position and the destination. As shown in Figure 3(a), the performance exhibits a decline as the hop count increases for both tasks. The initial rise in the Navigation task's SPL could potentially be attributed to the imposition of 4 to 6 hop constraints within our dataset, which encourages the model to learn the importance of not terminating prematurely, leading to longer TL and lower SPL at the beginning.

## 5.5 EMERGENT LANGUAGE ANALYSIS

One of crucial merits of language is to generate expressions grounded with world perceptions and mental activities. In order to investigate whether the emergent language holds such attribute, we conduct an analysis of the grounding between messages and visual observations. As shown in Figure 4(a), we depict the encoded observation vectors (points in orange), and their corresponding encoded message vectors (points in green) respectively in a reduced 2D coordinate space. We can observe that there are similar distributions of observation vectors and message vectors. To provide in-depth analysis, we cluster the observation vectors and randomly select 3 groups (A, B and C group) and show the connection lines (in different colors) between the observation vectors and their corresponding message vectors, from which we can observe that the same observation vector group corresponds to the same message cluster. We can also observe the parallels between lines connecting the message group centers and that connecting their corresponding observation group centers. For example, line a-c parallels to line d-f, line c-b parallels to line f-e and so on, proving that the message also successfully captures the inherent relationship within visual groundings. More cases in Figure 6 support these finding through observations that similar visual patterns share similar symbol n-grams. With those findings, we analyze that the emergent language could adhere to certain underlying rules, a topic we intend to explore in our forthcoming research.

## 6 CONCLUSION

The paper proposes and implements a collaborative multi-agent navigation via emergent dialogue, where the Tourist and the Guide learn expressions aligned with both the real-world environments and task-specific dialogue intents. The proposed methods achieve performance comparable to those of the established approach on the R2R and CVDN dataset, even without the need for human annotations or prior linguistic knowledge.

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

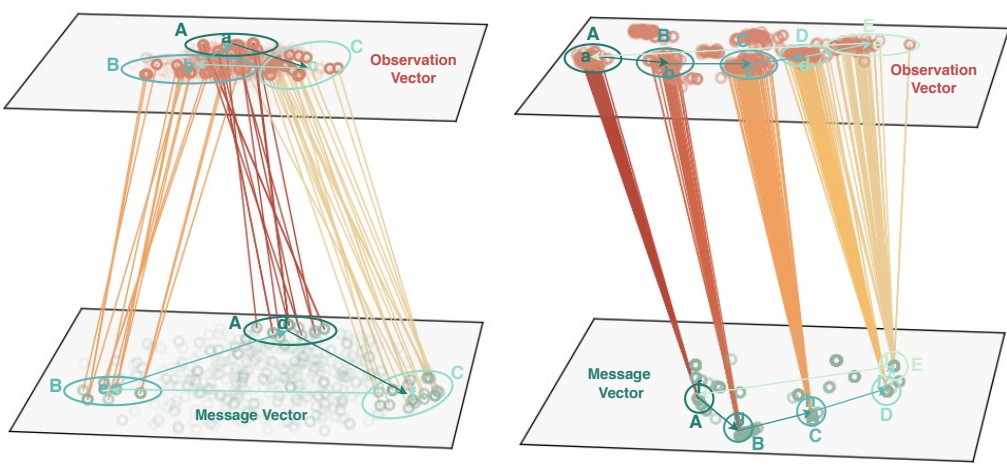

(a) Message and vision grounding.  (b) Message and action alignment.

Figure 4: Emergent language analysis. We apply PCA to map the message vectors and observation or action vectors to 2D space. Figure 4(a) depicts the distribution of the vectors and the relationship between observation vectors and its corresponding message vectors. Figure 4(b) shows the relationship between action vectors and its corresponding message vectors.

## A  FURTHER ANALYSIS OF EMERGENT LANGUAGE

To further explore the characteristics of the emergent language, we use the topographic similarity ($\rho$) to evaluate the compositionality of the language. This metric essentially measures the extent to which similar meanings correspond to similar languages. To compute $\rho$ in the Localizaton task, we first compute two lists of similarity scores: 1. The similarity score between all pairs of image observations. 2. The Levenshtein distances between corresponding pairs of messages. And the opposite of Spearman correlation between those two lists is the topographic similarity score.

To calculate the similarity score between image observations, we tried 3 types of methods: 1. The Euclidean distance between the image observation pairs. 2. The cosine distance between the feature vectors extracted by a pre-trained ResNet without training on navigation task or Matterport3D dataset. 3. The cosine distance between the feature vectors extracted by the environment encoder of the Tourist agent trained on this task. To calculate the Levenshtein distances between the message pairs, we tried 2 types of methods: 1. The Levenshtein distance between the utterances generated in the current turn. 2. The Levenshtein distance between the dialog histories up to the current turn.

The topographic similarity is on the order of 0.01 using methods 1 and 2 of calculating the similarity between observations, and is much higher using method 3, suggesting that the emergent language

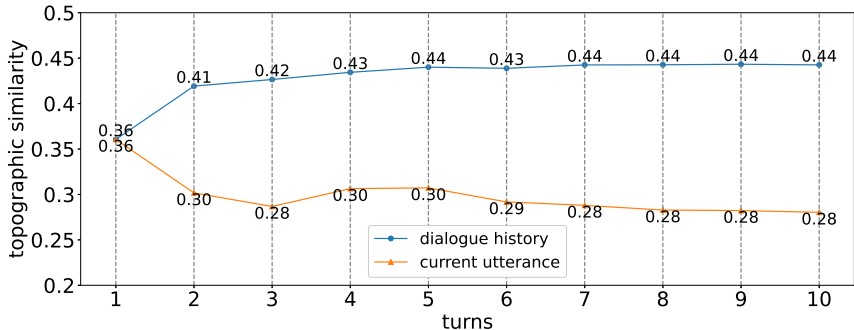

Figure 5: The topographic similarity $\rho$ change with dialogue turns. The blue line refers to the change of $\rho$ between the observation similarity list calculated by method 3 (cosine similarity between the task-related observation vectors) and the message similarity list computed by method 1 (The Levenshtein distance between the dialogue history up to current turn). The orange line refers to the change of $\rho$ between the observation similarity list calculated by method 3 and the message similarity list computed by method 2 (The Levenshtein distance between the current turn utterance).

captures different views about the surrounding and the task ontology with the natural language learned from a large scale of human-annotated data. Combining method 3 of observation similarity with two methods calculating message distance separately, the results show different trends as the number of dialogue rounds increases shown in Figure 5. The first setting observes an uptrend in topographic similarity (colored in orange), while the second setting is a downtrend (colored in blue). The uptrend suggests that the longer history conveys the more information. The downtrend suggests that each message progressively enriches information given the previous turns, and the messages at the beginning of the task carry more information than those of the later steps. As a metric evaluating the compositionality of the emergent language, this score suggests that the language could composite task-specific features helpful for task solving.

## B    BASELINE METHODS

Based on the number of modeled agents, we could divide the baselines into two categories: single agent Anderson et al. (2018); Tan et al. (2019); Zhao et al. (2022); Chen et al. (2021); Zheng et al. (2023); Wang et al. (2020), multiple agents Roman et al. (2020). The former methods focus on making the Tourist understand given instructions written by human, while the latter method not only requires both of the agents to understand language but also generate message to dialog with each other. Based on the method, according to Gu et al. (2022), we could divide the baselines into three categories: data-centric learning (DCL) Anderson et al. (2018); Tan et al. (2019), action strategy learning (ASL) Zhao et al. (2022), and representation learning methods (ReL) Chen et al. (2021); Zheng et al. (2023); Wang et al. (2020). Data-centric methods use the existing data effectively or create synthetic data. Strategy learning methods help agents to find the best actions. And representation learning methods enable the agents to understand the relationship between words and perceived features in the environment.

## C    DETAILS OF EMERGENT LANGUAGE ANALYSIS

In Figure 4(a) we use PCA as a dimension reduction technic to reduce the encoded Tourist's visual image $\bar{e}_o^t \in \mathbb{R}^{dim}$, and encoded dialog history $\bar{e}_{H_{dialog}}^G \in \mathbb{R}^{dim}$ into 2D space. Similarly, in Figure 4(b), the encoded dialogue history $\bar{e}_{H_{dialog}}^T \in \mathbb{R}^{dim}$, and the encoded next step action $\bar{e}_p^{pos_i \rightarrow pos_j} \in \mathbb{R}^{dim}$ are reduced via PCA dimension reduction tool. Then, in Figure 4(a), we cluster the encoded visual images using KMeans and connect the center of the clusters and the center of the corresponding messages, then we find the parallel. Similarly, in Figure 4(b), we cluster the encoded next step action using KMeans and connect the center of the clusters and the center of the corresponding messages.

---

**Algorithm 1:** Navigation

    **input** : Global environment $G$ and initial position of Tourist $pos_{t=0}$
    **output:** Dialogue history $H_{dialog}$ and navigation path $H_{act}$

**1**   **for** $t \leftarrow 1$ **to** $L^{turn}$ **do**
**2**      First, Tourist predict an action $a_t$ according to $H_{act}, A_t^T, H_{dialog}$;
**3**      $e_{H_{dialog}}^T \leftarrow DialEnc^T(H_{dialog})$;
**4**      $pred^{act}, e_{env}^T \leftarrow EnvEnc^T(o_t^T, H_{act}, A_t^T, e_{H_{dialog}}^T)$;
**5**      $a_t \leftarrow \arg\max pred^{act}$;
**6**      **if** $a_t ==$ *"stop"* **then** $pos_{t+1} \leftarrow pos_t$;
**7**      **else** $pos_{t+1} \leftarrow map(Neighbor(pos_t), a_t)$;
**8**      $H_{act} \leftarrow H_{act} + a_t$;
**9**      Second, Tourist generate an utterance $U_t^T$ according to $H_{dialog}, e_{env}^T$;
**10**      $e_t^T \leftarrow concat(e_{H_{dialog}}^T, e_{env}^T)$;
**11**      $U_t^T \leftarrow DialDec^T(e_t^T)$;
**12**      $H_{dialog} \leftarrow H_{dialog} + U_t^T$;
**13**      Third, Guide predict tourist position according to $V, H_{dialog}$;
**14**      $e_{H_{dialog}}^T \leftarrow DialEnc^G(H_{dialog})$;
**15**      $pred^{pos}, e_{env}^G \leftarrow EnvEnc^G(V, P, e_{H_{dialog}}),$;
**16**      Fourth, Guide generate an utterance $U_t^G$ according to $H_{dialog}, e_{env}^G$;
**17**      $e_t^G \leftarrow concat(e_{H_{dialog}}^G, e_{env}^G)$;
**18**      $U_t^G \leftarrow DialDec^G(e_t^G)$;
**19**      $H_{dialog} \leftarrow H_{dialog} + U_t^G$;
**20**      Update parameters according to equation 20;

---

## D    Procedure of Our Method

## E    Examples

Figure 6 shows the utterance and the observations of the Tourist in the localization task when the Tourist observes the environment from different angles but in the same positions. We can observe that the utterances are very similar which proves that the similar observations correspond to similar messages, but still there are differences between messages proving that there is observation angle information in the messages.

Although we can observe that the observation vectors in the same group tend to generate messages with vectors in a similar region, Figure 7 provides some cases in group C of Figure 4(a), making the conclusion much more intuitive.

| | | Conversation | GP |
|---|---|---|---|
| Human | 🔵 | Do I go in between the ropes to my right or straight forward? | 0 |
| | 🔴 | straight forward through the next room | |
| | 🔵 | Should I proceed down the hall to the left of turn right? | 13.31 |
| | 🔴 | head down the hall to your right into the next room | |
| | 🔵 | Should I go through the open doors that are the closest to me? | 29.05 |
| | 🔴 | You are in the goal room | |

| | | Conversation | GP | | | Conversation | GP |
|---|---|---|---|---|---|---|---|
| RMM | 🔵 | should i head forward or bedroom the next hallway in front of me ? | 4.0 | Ours | 🔵 | j i j j j j j j j j j j | 0.0 |
| | 🔴 | yes , all the way down the small hall . | | | 🔴 | t k t t t t t t t t t t | |
| | 🔵 | should i turn left here ? | 5.7 | | 🔵 | r e i g v e e e e e e e | 0.0 |
| | 🔴 | head into the house , then you will find a doorway at the goal staircase . go through the doors before those two small exit chairs , about half way down the hall . | | | 🔴 | t k t t t t t t t t t t | |
| | 🔵 | lots of sink in this house , or wrong did . ok which way do i go | 8.8 | | 🔵 | r f b r r r r r r j r i | 2.19 |
| | 🔴 | go down the hallway , take a left and go down the next hallway and up the stairs on the right . | | | 🔴 | k k k k k k k k k k k k | |
| | 🔵 | right place now ? | 7.5 | | 🔵 | f s b r j s e b s v b s | 3.95 |
| | 🔴 | turn left and when you reach to the bathroom . before these go up the stairs on your left , and walk passed the mirror and chairs on the left into hallway across from the stairs | | | 🔴 | l t l t l t l t l t l t | |
| | 🔵 | do i go left , right , or up the stairs ? | 4.0 | | 🔵 | f e z g j f z b e i j e | 5.69 |
| | 🔴 | turn to your left and keep going straight | | | 🔴 | k k k k k k k k k k k k | |
| | 🔵 | do i make a right turn or shall i enter the room to my left ? | 0.4 | | 🔵 | r e x z x g b e b e b e | 7.52 |
| | 🔴 | turn left to go down the hall , then that fire bed has that | | | 🔴 | t k t t t t t t t t t t | |
| | 🔵 | left or continue ? | 0 | | 🔵 | x b z i g x i x i x i x | 8.78 |
| | 🔴 | yes , go past the dining table and take an immediate right . head through the small door to the left of the window before those two way doors behind you , go up the second small set of stairs . | | | 🔴 | k k k k k k k k k k k k | |
| | 🔵 | should i go downstairs or turn left ? | 4.0 | | 🔵 | f g v r j g g g g g g g | 9.85 |
| | 🔴 | go to the right and go to the front doorway . | | | 🔴 | l k l k l k l k l k l k | |
| | 🔵 | should i go down the stairs or stay on this floor ? | 8.8 | | 🔵 | f e s x i s e j s i s i | 12.18 |
| | 🔴 | take a left towards the bathroom , then take a left before it and go all the way down the hall | | | 🔴 | t l t l t l t l t l t l | |
| | 🔵 | do i go up these is to the right or right of the steps ? | 9.9 | | 🔵 | f e s j s e i e f s e e | 15.93 |
| | 🔴 | go to the left side of the staircase and turn left in the doorway before the two small office chairs , about half way down the hall . | | | 🔴 | l t l t l t l t l t l t | |
| | 🔵 | should i turn left , go straight into the living room , or up the stairs ? | 7.5 | | 🔵 | e g s e j e e e e e e e | 18.62 |
| | 🔴 | turn to your right and go straight down the hall | | | 🔴 | t y t y t y t y t y t y | |
| | 🔵 | do i go out into the hallway ? | 5.7 | | 🔵 | e g s e j g g g g g g g | 21.02 |
| | 🔴 | go left down the hall where the office floor . and pass the second door after a right and table . | | | 🔴 | t l t l t l t l t l t l | |
| | 🔵 | ok , should i go right or left next ? | 8.8 | | 🔵 | e g s e j g g g g g g g | 21.02 |
| | 🔴 | go back to the staircase . go through the doorway you and before the hallway on the right . | | | 🔴 | t l t l t l t l t l t l | |
| | 🔵 | do i make a left turn or shall i enter the room to my left ? | 13.3 | | 🔵 | f s v s v g s r s e s b | 21.02 |
| | 🔴 | go down the hall and turn right into the bedroom | | | 🔴 | l t l t l t l t l t l t | |
| | 🔵 | should i go to the left or the right ? | 9.3 | | 🔵 | f s v s v g s r s e s b | 23.82 |
| | 🔴 | yes , go out of this room , turn right and go down the white hall before the staircase stairs , then go down the way down that way you get . | | | 🔴 | l g l g l g l g l g l g | |
| | 🔵 | ok i was a in by this office painting , or i just in the second hallway in front of me ? | 9.3 | | 🔵 | e s g j e e e e e e e e | 23.82 |
| | 🔴 | okay . | | | 🔴 | l t l t l t l t l t l t | |
| | 🔵 | which way do i go in , or do i head up the stairs ? | 11.1 | | 🔵 | f s v s v g s r s e s b | 21.02 |
| | 🔴 | go all the way to the one of the staircase . turn left in the doorway before the two two office chairs , about half way down the hall . | | | 🔴 | l g l g l g l g l g l g | |
| | 🔵 | ok wrong far which way do i go | 8.8 | | 🔵 | e s g j e e e e e e e e | 23.82 |
| | 🔴 | right then at the top of the stairs . | | | 🔴 | l t l t l t l t l t l t | |
| | 🔵 | left or continue ? | 7.5 | | 🔵 | f s v s v g s r s e s b | 21.02 |
| | 🔴 | yes . go down the hall and stop at the landing of the stairs . | | | 🔴 | l g l g l g l g l g l g | |

Table 5: Conversation Example

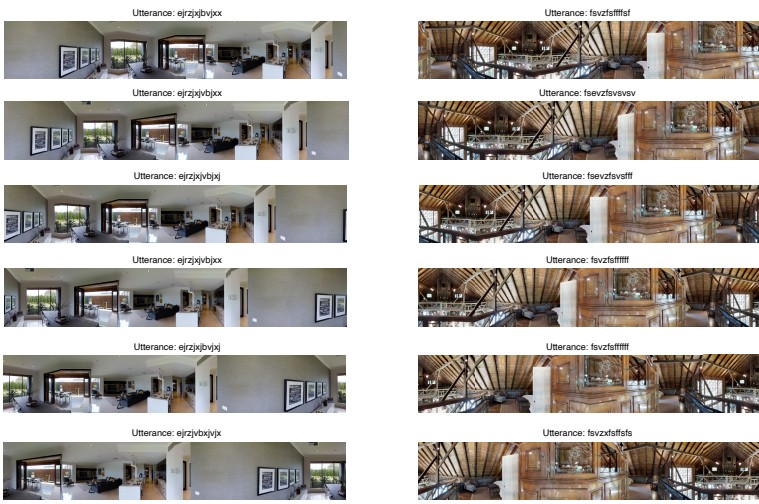

Figure 6: The first utterance of the Tourist in the localization task when the Tourist observe the environment from different angles of the same position.

Utterance: xsevbxxxxxxx

Utterance: xjisjxxxxxxx

Utterance: xvfxxxxzixxx

Utterance: xeizxxxxfxsi

Figure 7: Cases in Group C in Figure 4(a). We can observe that similar observation corresponds to similar messages

