# OpenReview forum: "Emergent Language based Dialog for Collaborative Multi-agent Navigation"
_ICLR.cc/2024/Conference — Submitted to ICLR 2024_

### Official Review · Reviewer_1ZPo · 2023-10-31

**Soundness:** 2 fair
**Presentation:** 3 good
**Contribution:** 2 fair
**Rating:** 5
**Confidence:** 2

**Summary:**

This paper proposes a collaborative multi-agent navigation via emergent dialogue, where the Tourist and the Guide learn expressions aligned with both the real-world environments and task-specific dialogue intents. The proposed method enables both agents to generate and understand emergent language, and develop optimal dialogue decisions with a long-term goal of solving the task. The paper provides a real-world navigation scene with matterport3D simulator to showcase the effectiveness of the proposed method.

**Strengths:**

1. The authors propose a novel multi-agent reinforcement learning (RL) framework complemented by auxiliary pre-training to effectively align emergent language with both the environment and the task.
2. Experimental results on a real-world navigation scene with matterport3D simulator to showcase the effectiveness of the proposed method.

**Weaknesses:**

1. The paper does not provide a detailed analysis of the underlying rules that the emergent language adheres to, which may limit the understanding of the method.
2. The design of each module of the method is relatively conventional, and no particular contribution was found.

**Questions:**

Can you provide some text examples to compare the differences between the text learned from the best baseline and the method learned from your own method?
Can you explain again what advantages the design of each module in the method has compared to the previous method?

---

> ### Author Response · Authors · 2023-11-21
>
> We thank the reviewer for the valuable comments, and hope the following point-by-point responses address the reviewer’s concern.
>
> Question 1:
>
> Among the mentioned baselines, RMM [1] is the only work concerning language generation ability. Unfortunately, it’s hard to reproduce RMM to get sufficient text examples for comparison. We take the example provided in the RMM paper, run our model using the same start position and the same target, and compare the emergent texts of our model with the natural language of RMM, as depicted in Appendix Table 5. There is no obvious alignment between the natural language and the emergent language. This observation draws a similar conclusion as that of Appendix A, i.e., the emergent language captures different views about the surrounding and the task ontology with the natural language learned from a large scale of human-annotated data. This will lead to an interesting content of our future work.
>
> Question 2:
>
> Much existing research on navigation task concerns only language understanding ability of one agent. The most similar work is RMM [1], which focuses on training the natural language generation and understanding of two agents, under the supervision of a large scale of labelled dialogues. While our work concerns about the emergent language generation and understanding, without any labelled language data. The intrinsic advantage of our design is to train a multi-agent navigation model all from the scratch, without any human intervention, which is, to the best of our knowledge, the first work that proves the feasibility of multi-pass emergent communication in realistic dynamic environments, and provides an empirical method for the implementation.
>
> Weakness 1:
>
> Thank you for pointing out this concern. There is no explicit underlying rules or heuristics that the emergent language adheres to. Actually, we performed intensive analysis of the emergent language, by detecting the alignment of emergent language with visual surrounding, and of emergent language with task, which we consider as two essential characters of language. In order to make further study on whether there are implicit rules underlying the emergent language, following [2], we analyzed the compositionality of emergent language, by calculating the topographic similarity between observation pairs and corresponding message pairs under two different settings.
>
> In the first setting, we calculate the editing distances of two dialogue histories up to the same turn $t$ as following:
>
> $$LevenshteinDistance(H_{utterance, t}^{T, i}, H_{utterance, t}^{T, j})$$
>
> $$H_{utterance, t}^{T, i} = \langle U_1^{T, i}, U_2^{T, i}, ..., U_t^{T, i}\rangle$$
>
> $$H_{utterance, t}^{T, j} = \langle U_1^{T, j}, U_2^{T, j}, ..., U_t^{T, j}\rangle$$
>
> In the second setting, we calculate the editing distances of two messages in the same turn $t$:
>
> $$LevenshteinDistance(U_t^{T, i}, U_t^{T, j})$$
>
> The curves of topographic similarity are shown in the Figure 5 in the paper. The first setting observes an uptrend in topographic similarity (colored in orange), while the second setting a downtrend (colored in blue). The uptrend suggests that the longer history conveys the more information. The downtrend suggests that each message progressively enriches information given the previous turns, and the messages at the beginning of the task carry more information than that of the later steps. As a metric evaluating the compositionality of the emergent language, this score suggests that the emergent language could composite task-specific features helpful for task solving.
>
> Weakness 2:
>
> Please refer to the response to Question 2.
>
> [1] Roman H R, Bisk Y, Thomason J, et al., RMM: A recursive mental model for dialogue navigation, EMNLP 2020.
>
> [2] Lazaridou A, Hermann K M, Tuyls K, et al., Emergence of linguistic communication from referential games with symbolic and pixel input, ICLR 2018

---

### Official Review · Reviewer_YMT3 · 2023-10-31

**Soundness:** 3 good
**Presentation:** 3 good
**Contribution:** 3 good
**Rating:** 6
**Confidence:** 4

**Summary:**

In this paper, the authors introduce a model of communication between two agents, a Tourist and a Guide. The former must navigate an environment to reach a unknown target location following instructions provided by the latter. The Guide, without knowing the current position of the Tourist, must communicate to provide information about the path to the target position. Using two objective functions, one optimizing for Guide localization of the Tourist and one for optimizing for Tourist navigation, they report better results over previous work on two visual-language navigation datasets.

**Strengths:**

I think the authors investigate the interesting problem of embodies agents communicating while having different roles, a follower and a guide. The lack of any human-annotation in the training process makes it for cheap method to train embodies agents and the ablation study conducted supports the modelling choices made by the authors. Furthermore, the usage of realistic datasets shows that the method can scale to natural setups and is an important step towards deploying agents in the wild.

Overall, I find the only weaknesses to be in the lack of an in-depth analysis of the emergent language. (more in the following section) The only analysis shown is qualitative one and is relegated to the appendix. Although, I don't see it a fundamental requirement to accept the paper, what prevented me from giving it a higher score is the lack of a deeper analysis.

Finally, provided a minor restructuring of the manuscript is done, such as moving the analysis to the main body of the paper, I am in favor of including the paper at the conference.

**Weaknesses:**

As I mentioned in the previous section, I think the major weakness of the paper is in the lack of deep analysis of the emergent language. Computing metrics of emergent languages like topographic similarity (using environment encodings and agent messages) [1] could give an idea of the structure of the agents' protocols.

A minor weakness that I found is the usage of agents without any pre-encoded linguistic knowledge. Using a LLM as navigation planner, which has shown some potential [2, 3], could solve the problem of training agents that have an opaque communication protocol.
I am thinking that your method could be used as a fine-tuning approach over existing language-aware models, keeping in mind that the language drift problem [4] should be taken into account. I'd be happy to hear the authors' opinion on this


[1] Lazaridou et al.,  Emergence of linguistic communication from referential games with symbolic and pixel input, ICLR 2018

[2] Rana et al, SayPlan: Grounding Large Language Models using 3D Scene Graphs for Scalable Robot Task Planning, CoRL 2023

[3] Rajvanshi et al., SayNav: Grounding Large Language Models for Dynamic Planning to Navigation in New Environments, 2023

[4] Lazardou et al., Multi-agent communication meets natural language: Synergies between functional and structural language learning, ACL 2020

**Questions:**

- At the end of the related work section you claim: "we also empower the oracle agent to provide instructions progressively", how is it guaranteed that messages are sent progressively and not all at the beginning? While I'm not challenging your claim, I'm wondering whether it's backed by any analysis work or just by the nature of your sequential communication modules.

- In the training setup section I don't understand why you call the two objective "pre-training" tasks. Aren't they used jointly to train the agents? From the paper, I don't understand, the division, if any, between pre-training and training.

- Why do you choose a vocabulary size of 26? I first thought it was to draw a similarity with the English language, but I then realized by looking a figure 3b that it could be the result of an hyperparameter optimization following an ablation study. Could you please clarify?

- In sec 5.5, how do you compute the reduced 2D space? Could please you provide additional details? They could easily be added to the appendix for a camera-ready version


---------

misc/typos

- In section 4.1 you mention Guest position, do you mean Tourist position?

- In the related work section you mention: "[...] has a similar setting to our work but lets the Guide describe the target position’s observation in a kind of emergent language". I find "a kind of emergent language" unclear, please fix it.

- Please provide a more descriptive Figure 4 caption than the rather vague "Emergent language analysis"

- "Language based" in the title is probably missing a "-" -> Language-based

Another related paper about navigation and emergent communication is [1]. Despite their communication modules being simpler than your, I think it makes sense to include it in your section surveying the literature.

[1] Patel et al., Interpretation of Emergent Communication in Heterogeneous Collaborative Embodied Agents, ICCV 2021.

---

> ### Author Response · Authors · 2023-11-20
>
> We thank the reviewer for the valuable and constructive comments, and add the following clarifications to address the reviewer’s concerns:
>
> Question 1:
>
> The claim “we also empower the oracle agent to provide instructions progressively “is backed by the experiment results on different settings of dialogue strategy which is shown in Table 4. When comparing the “only in beginning” setting with other settings with more dialogue frequency, it observes that the “only in. beginning” setting has a poorer performance, indicating that providing all information only at the beginning is not enough, and the message in each turn conveys additional helpful information for the following task. To better verify this point, we conducted experiments of “direct communication” (i.e., sending the shortest path repeatedly in each turn) and observed a similar performance with that of the “only in beginning” setting (tl=10.09, ne=6.88, sr=25.74, spl=22.78, vs tl=10.79, ne=6.44, sr=25.37, spl=21.13), proving that the subsequent utterances provide additional helpful information instead of repeating the same information. Our further study also found that especially when the Tourist made wrong movement actions, the multi-turn communication could provide helpful instructions, leading to a much better performance.
>
> Question 2:
>
> At the very beginning of our research, we tried to directly train a multi-agent navigation model in an end-to-end manner using reinforcement learning, and observed a discouraging non-convergence of the training. Under this setting, we jointly trained the language generation module and understanding module for both agents (i.e., four modules in total) by simultaneously maximizing the whole accumulated rewards. However, the model collapsed into a poor status since the generation and understanding are not coordinated at the start of training which can cause the state to quickly become large and chaotic. Therefore, we formulate the navigation task as a two-pass task and train each pass separately in a supervised learning manner: In the first pass (the so-called “Localization” task), the Tourist was encouraged to learn how to generate language accurately describing its visual surrounding, and the Guide to correctly understand the message from the Tourist to predict where the Tourist is, with the goal of making proper guideline for future steps. In the second pass (the so-called “Movement” task), the Guide was encouraged to generate language accurately describing its guideline, and the Tourist to correctly understand the message conveying the guideline, in order to make proper movement. Then we returned back to jointly training for the navigation task as mentioned at the beginning, but with parameters initialized by the two-pass training tasks. Therefore, we call each pass a pre-training task of the targeted navigation task, the principled fundamentals of which are explained in Section 4.5.
>
> Question 3:
>
>  Yes, your thought is right. We do choose the vocabulary size of 26 to draw a similarity with the English language. The experiment in Figure 3b is to explore the effect of the vocabulary size on the performance and the 26 may not be the best choice among all possible choices of vocabulary size in the proposed navigation task.
>
> Question 4:
>
> In Figure 4 (a) of section 5.5, we use PCA as a dimension reduction technique to reduce the encoded Tourist’s visual image $\bar{e} _o^t \in \mathbb{R}^{dim}$, and encoded dialog history $\bar{e} _{H _{dialog}}^G \in \mathbb{R}^{dim}$ into 2D space. Similarly, in Figure 4 (b), the encoded dialogue history $\bar{e} _{H _{dialog}}^T \in \mathbb{R}^{dim}$, and the encoded next step action $\bar{e} _p^{pos _i \rightarrow pos _j } \in \mathbb{R}^{dim}$ are reduced via PCA dimension reduction tool. Then, in Figure 4 (a), we cluster the encoded visual images using KMeans and connect the center of the clusters and the center of the corresponding messages, then we find the parallel. Similarly, in Figure 4 (b), we cluster the encoded next step action using KMeans and connect the center of the clusters and the center of the corresponding messages. Thanks for your advice, and we have added those details to the appendix.

---

> ### Author Response · Authors · 2023-11-20
>
> Weakness 1:
>
> Thanks very much for the suggestion, inspired by your suggestions we have conducted some analysis about the emergent language, and have added it to the paper. More efforts will be made to have a deeper analysis of the language.
>
> we conducted an experiment to analyze the compositionality of emergent language, by calculating the topographic similarity between observation pairs and corresponding message pairs under two different settings.
>
> In the first setting, we calculate the Levenshtein distances of two dialogue histories generated by the Tourist up to the same turn $t$.
>
> $$LevenshteinDistance(H _{utterance, t}^{T, i}, H _{utterance, t}^{T, j})$$
>
> $$H _{utterance, t}^{T, i} = \langle U _1^{T, i}, U _2^{T, i}, ..., U _t^{T, i}\rangle$$
>
> $$H _{utterance, t}^{T, j} = \langle U _1^{T, j}, U _2^{T, j}, ..., U _t^{T, j}\rangle$$
>
> In the second setting, we calculate the Levenshtein distances of two messages in the same turn $t$.
>
> $$LevenshteinDistance(U _t^{T, i}, U _t^{T, j})$$
>
> The curves of Topographic similarity are shown in Figure 5 in the paper. The first setting observes an uptrend in topographic similarity (colored in orange), while the second setting is a downtrend (colored in blue). The uptrend suggests that the longer history conveys the more information. The downtrend suggests that each message progressively enriches information given the previous turns, and the messages at the beginning of the task carry more information than those of the later steps. As a metric evaluating the compositionality of the emergent language, this score suggests that the language could composite task-specific features helpful for task solving.
>
> Those results and more details have been added to the paper and we are trying hard to explore more characteristics of the emergent language.
>
> Weakness 2:
>
> It is a very interesting idea to apply our method to fine-tune a language-aware model. Since the proposed method is formulated for navigation tasks, the emergent language could capture the most task-related information directly. By comparison, training both of the agents (the Tourist and the Guide) learning natural language, requires the language to capture two types of information: the distribution of natural language and the task-specific information. In natural language setup, a direct method is to take a supervised method to train the language modules to achieve the task. However, in supervised training, the quality of natural language is learned directly but the task-related information lays behind the language which is learned indirectly. This is a possible reason why the natural language has a poorer performance (proved by the comparison between our method with the RMM). Applying our method to a language-aware model could mitigate this problem. With our method learning the task-related information, the supervised method learning the language structure, a much better performance could be achieved.
>
> Some possible methods to incorporate our method into the existing language-aware model without causing much language drift could be tried:
>
> 1. Use our method as a finetuning approach with KL loss making sure that the fine-tuned model is not too far from the initial language-aware model, similar to the RLHF methods applied on LLM.
> 2. Train the supervised method and our RL methods on the language modules iteratively. The supervised method is used to maintain the quality of natural language, and our RL method to learn task-related information following [1].
>
>
> We apologize for the misc/typos, and have fixed them and also added the missing reference. We will also move Figure 4 (Emergent language analysis) to the main body of the paper in the revised version.
>
>
> [1] Lowe R, Gupta A, Foerster J, et al., On the interaction between supervision and self-play in emergent communication, ICLR 2020.

---

### Official Review · Reviewer_cd7K · 2023-11-01

**Soundness:** 3 good
**Presentation:** 2 fair
**Contribution:** 4 excellent
**Rating:** 5
**Confidence:** 3

**Summary:**

This study introduces a multi-agent navigation task in which one agent, the Tourist, perceives its immediate surroundings, while the other agent, the Guide, has an overarching view of the environment but lacks knowledge of the Tourist's location. The primary objective is for the Guide to direct the Tourist to a specific destination using evolving multi-turn dialogues. The paper details an empirical study centered on developing agents that can collaborate effectively through multi-turn emergent dialogues. To exemplify this, the authors introduce a collaborative multi-agent reinforcement learning technique that enables both agents to generate and understand emergent language. This method is tested with the Matterport3D simulator.

**Strengths:**

1. The paper introduces a novel framework for emergent communication based on a vision-language navigation task.
1. An empirical study is provided as an example. That will be used as a baseline in future studies.
1. The reviewer believes that the task design will contribute significantly to expanding the study of emergent communication.

**Weaknesses:**

1. The paper does not sufficiently assess the quality and characteristics of the emergent language, such as its compositionality and its relation to the plan set out by the Guide.
1. The figures contain very small text, making them hard to understand.
1. While the main contribution seems to be the proposal of the task, a large portion of the description is dedicated to the network architecture (Section 3). The authors should provide a more intuitive explanation of the general task framework. Adding pseudocode could help potential readers grasp the proposal more effectively.
1, The characteristics and details of the compared baseline methods in the experiments are not clear. It would be beneficial to include these descriptions in the Appendix.

**Questions:**

The most straightforward communication approach the Guide could adopt is to repeatedly send the shortest path. How does the emergent language compare with such direct communication? If this hasn't been explored, it would be worthwhile to discuss.

---

> ### Author Response · Authors · 2023-11-19
>
> We thank the reviewer for the valuable comments, and hope the following clarification addresses the reviewer’s concern.
>
> Questions:
>
> Thank you for your thought-provoking question. We had conducted experiments in a similar setting, referring to “only in beginning” in Table 4. We let the Tourist encode the shortest path instruction as a part of dialog history in each turn. Inspired by your question, we immediately conducted experiments using “direct communication” (i.e., sending the shortest path repeatedly in each turn) and observed a similar performance with that of the “only in beginning” setting (tl= 10.09, ne=6.88, sr=25.74, spl=22.78, vs tl = 10.79, ne=6.44, sr=25.37, spl=21.13). By comparing with other results in Table 4, both performance of “direct communication” and “only in beginning” are poorer than that under other settings (“soft”, “20%”,  “50%”,  “100%”), suggesting that the dialogue progressively provides additional information. Our further study also found that especially when the Tourist made wrong movement actions, the multi-turn communication could provide helpful instructions, leading to a much better performance.
>
> Weakness 1:
>
> Thank you for your suggestion, and we had noticed this problem before but did not make much attempt at it. We performed language analysis by detecting the alignment of emergent language with visual surroundings, and that of emergent language with task, which we consider as the essential characters of language. Motivated by your suggestion, we conducted an experiment to analyze the compositionality of emergent language, by calculating the topographic similarity between observation pairs and corresponding message pairs under two different settings.
>
> In the first setting, we calculate the Levenshtein distances of two dialogue histories generated by the Tourist up to the same turn $t$.
>
> $$LevenshteinDistance(H_{utter, t}^{T, i}, H_{utter, t}^{T, j})$$
>
> $$H_{utter, t}^{T, i} = \langle U_1^{T, i}, U_2^{T, i}, ..., U_t^{T, i}\rangle$$
>
> $$H_{utter, t}^{T, j} = \langle U_1^{T, j}, U_2^{T, j}, ..., U_t^{T, j}\rangle$$
>
> In the second setting, we calculate the Levenshtein distances of two messages in the same turn $t$.
>
> $$LevenshteinDistance(U_t^{T, i}, U_t^{T, j})$$
>
> The curves of Topographic similarity are shown in the Figure 5 of the paper. The first setting observes an uptrend in topographic similarity (colored in orange), while the second setting is a downtrend (colored in blue). The uptrend suggests that the longer history conveys the more information. The downtrend suggests that each message progressively enriches information given the previous turns, and the messages at the beginning of the task carry more information than those of the later steps. As a metric evaluating the compositionality of the emergent language, this score suggests that the language could composite task-specific features helpful for task solving.
>
> Those results and more details have been added to the paper and we are trying hard to explore more characteristics of the emergent language.
>
> Weakness 2:
>
> Thanks for your suggestion. We do find the font size is too small when we zoom in, which is unfriendly to readers. We have adjusted (and further improvements will be made) Figures 2 & 3 carefully and hope it could improve the readability of this paper.
>
> Weakness 3:
>
> Thank you for your suggestion and the old version is indeed not clear enough. We have added the pseudocode and the descriptions of the baseline methods to the Appendix (and more improvements will be made). And hope the revised version could be more friendly to readers.

---

> > ### Comment · Reviewer_cd7K · 2023-11-21
> > **Thanks**
> >
> > Thanks for your clarification.  I hope the revision has made the paper more productive and inspiring.
> > I am almost satisfied with your update.
> >
> > >We had conducted experiments in a similar setting, referring to “only in beginning” in Table 4. We let the Tourist encode the shortest path instruction as a part of dialog history in each turn. Inspired by your question, we immediately conducted experiments using “direct communication” (i.e., sending the shortest path repeatedly in each turn) and observed a similar performance with that of the “only in beginning” setting
> >
> > Considering that potential readers may have a similar question, it would be better to explicitly describe the similarity between "only in the beginning" and "direct communication" in the body or in a footnote.
> >
> > My score has been updated based on your response and revision.

---

> > > ### Author Response · Authors · 2023-11-22
> > >
> > > Thanks very much for your recognition and valuable suggestions. We will add the explanation to the main body in the revised version.

---

### Official Review · Reviewer_oveQ · 2023-11-03

**Soundness:** 3 good
**Presentation:** 2 fair
**Contribution:** 2 fair
**Rating:** 5
**Confidence:** 3

**Summary:**

This paper studies how to build agents that can collaborate effectively with multi-turn emergent dialogues, it proposes a multi-agent navigation task, to guide the Tourist (the agent) to reach the target place via multi-turn dialogues. It proposes a collaborative multi-agent reinforcement learning method that enables both agents to generate and understand language, and make decisions with a long-term goal of solving the task. Empirical experiments on R2R and CVDN tasks show promising results.

**Strengths:**

1. It introduces a multi-turn dialog for goal oriented navigation tasks.
2. It proposes a multi-agent RL algorithm for the task, and shows promising results on two tasks (R2R and CVDN).

**Weaknesses:**

Based on the motivation of this work, CVDN is a natural task for this method. It is better to show the performance on the test splits (unseen) comparing with SoTA methods, rather than only showing the val split.

**Questions:**

1. Is it possible to show the results for Test Unseen on R2R? besides the results on val seen and unseen. Similarly for CVDN dataset.

Minor suggestions:
1. Figure 2 & 3, the font is too small to read.

---

> ### Author Response · Authors · 2023-11-19
>
> We thank the reviewer for the valuable comments, and hope the following clarification addresses the reviewer’s concern.
>
> Weaknesses:
> we thank the reviewer for giving us valuable advice. The R2R and CVDN datasets are initially designed for natural language navigation between two agents, therefore their test data are used to evaluate the alignment quality of natural language with the environment, which are demonstrated on the corresponding leaderboards (but with the environment settings unknown to the public). Our work focuses on emergent language communication therefore it is infeasible to draw a comparison with related work on the leaderboard using test data. However, in order to assess the language understanding and generation ability of our method, we train and evaluate it under paralleled environment settings (i.e., with the same visual groundings of train and val dataset) with the related work on R2R and CVDN. We have mentioned this in section 5.1 and will give more explanations in the future revised version.
>
> Question1:
> Please refer to the above response to Weaknesses.
>
>
> Question 2:
> Thanks for your suggestion. We do find the font size is too small when we zoom in, which is unfriendly to readers. We have adjusted (and further improvements will be made) Figures 2 & 3 carefully and hope it could improve the readability of this paper.

---

### Meta-Review · Area_Chair_MbPs · 2023-12-06

**Metareview:**

Generally, the reviewers and the AC are not sure what to take from this paper. It has no implications on the natural language problem. Indeed, it does not aim to study it. The techniques and architectures are fairly customized, which means the qualities of the language learned provides little insight into broadly used techniques and models. The reviewers raised these concerns in different ways. The authors attempted to add analysis in the response, and potentially expanding this (beyond what we is possible within the short response window) can address some of the issues the reviewers raised. However, it's not clear it will address the fundamental issues of what is the general important question that is answered here. The properties that arise in an emergent language under a specific architecture, learning algorithm, and task seems very specific. How do they generalize?

Furthermore, I recommend the authors to revisit their decision to use R2R (including its trajectories!). This dataset has been shown to present issues. It was replaced by RxR, which addresses many of these issues and uses the same environment.

**Justification For Why Not Higher Score:**

See above.

**Justification For Why Not Lower Score:**

N/A

---

### Decision · Program_Chairs · 2024-01-16

Reject